# Ref-Adv: Exploring MLLM Visual Reasoning in Referring Expression Tasks

**Qihua Dong    Kuo Yang    Lin Ju    Handong Zhao    Yitian Zhang    Yizhou Wang**
**Huimin Zeng    Jianglin Lu    Yun Fu**

Northeastern University

## ABSTRACT

Referring Expression Comprehension (REC) links language to region level visual perception. Standard benchmarks (RefCOCO, RefCOCO+, RefCOCOg) have progressed rapidly with multimodal LLMs but remain weak tests of visual reasoning and grounding: (i) many expressions are very short, leaving little reasoning demand; (ii) images often contain few distractors, making the target easy to find; and (iii) redundant descriptors enable shortcut solutions that bypass genuine text understanding and visual reasoning. We introduce Ref-Adv, a modern REC benchmark that suppresses shortcuts by pairing linguistically nontrivial expressions with only the information necessary to uniquely identify the target. The dataset contains referring expressions on real images, curated with hard distractors and annotated with reasoning facets including negation. We conduct comprehensive ablations (word order perturbations and descriptor deletion sufficiency) to show that solving Ref-Adv requires reasoning beyond simple cues, and we evaluate a broad suite of contemporary multimodal LLMs on Ref-Adv. Despite strong results on RefCOCO, RefCOCO+, and RefCOCOg, models drop markedly on Ref-Adv, revealing reliance on shortcuts and gaps in visual reasoning and grounding. We provide an in depth failure analysis and aim for Ref-Adv to guide future work on visual reasoning and grounding in MLLMs. The dataset is available at https://ref-adv.github.io/.

## 1 INTRODUCTION

Referring expression comprehension (REC) is the task of grounding a natural language expression to a specific region in an image (Mao et al., 2016; Kazemzadeh et al., 2014; Yu et al., 2016). It has important applications in real world systems and downstream tasks, and it has become a key benchmark for evaluating multimodal large language models (MLLMs) because it probes fine grained correspondence between language and vision. Recent MLLMs (Google, 2025a; Bai et al., 2025; Zhu et al., 2025), both closed source and open source, have made substantial progress, achieving over 90% accuracy on classic REC benchmarks, i.e., RefCOCO(+/g) (Kazemzadeh et al., 2014; Yu et al., 2016; Mao et al., 2016).

Despite this near saturated performance, we identify critical limitations of the classic REC benchmarks that motivate a modern benchmark capable of more challenging and comprehensive evaluation of MLLMs. We view modern REC for MLLMs as a multistep reasoning task with two coupled components: (1) textual reasoning—understanding the referring expression, identifying the target, and identifying its descriptors; and (2) visual reasoning—searching for candidates and establishing correspondence between descriptors and image regions. The order of these steps can vary across models, but a meaningful benchmark should require both textual and visual reasoning. From this perspective, we highlight the following limitations of RefCOCO(+/g).

First, **_most of the referring expressions are extremely short_**, as shown in Figure 1. For RefCOCO and RefCOCO+, the average expression length is around 3 words. Such short expressions lead to two issues: (1) minimal linguistic effort is required, and (2) they typically entail less visual reasoning because fewer descriptors must be verified in the image. Second, **_there are few distractors in the images in RefCOCO(+/g)_**, as shown in Figure 2 (b), with most cases of only 1 distractors. Here we define a distractor as an object of the same category as the target but a different instance. When few

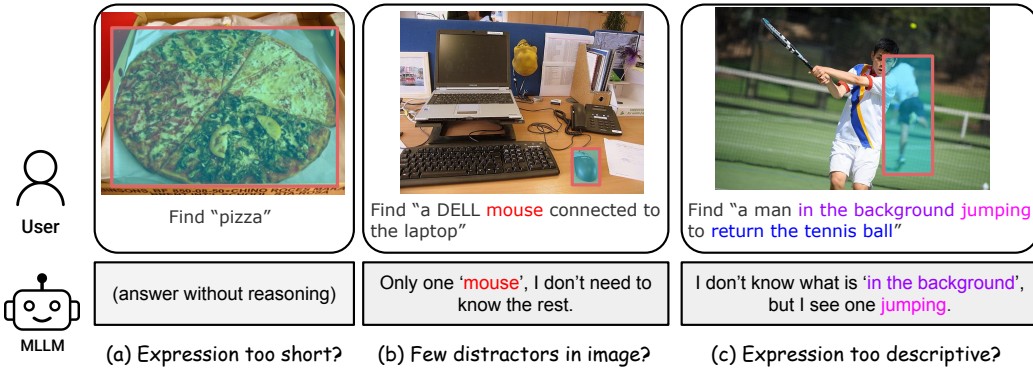

Figure 1: Common limitations of classic referring expression benchmarks that reduce the reasoning challenge. These include very short expressions, few visual distractors, and overspecified descriptors that enable shortcut matching without requiring genuine reasoning. The cyan box highlights the ground truth region.

distractors exist, the task requires far less textual and visual reasoning: models need only infer the target category and select from a small set of candidates. Figure 2 (b) reveals a negative correlation between the number of distractors and model performance.

It is worth noting that for reasoning assessment, *task difficulty does not monotonically increase with referring expression length due to "grounding shortcuts"*. These shortcuts occur when a long, descriptive expression is paired with few distractors, rendering many descriptors redundant. Consequently, a model can localize the target by matching only a subset of descriptors, which can paradoxically lead to higher accuracy for longer expressions, as illustrated in Figure 2 (a). This highlights the need for modern REC benchmarks to mitigate such shortcuts by designing expressions that are concise and carefully balanced against the available distractors.

Meanwhile, prior work has acknowledged aspects of these limitations: Wei et al. (2024); Chen et al. (2024) point out the length limitations of RefCOCO(+/g), and Chen et al. (2020) highlights the lack of distractors. However, the proposed datasets also raise new concerns. The former introduces REC data with average length $\geq 90$ words, which may be unnatural and, more importantly, enable numerous shortcuts since the numbers of descriptors and distractors are heavily imbalanced. The latter proposes settings including referring from a set of images, which shifts away from the classic REC setting, and the referring expressions are sampled from GQA (Hudson & Manning, 2019) scene graphs with fixed templates, reducing naturalness.

We therefore aim to build a REC benchmark that preserves the classic REC setting and natural expressions while substantially increasing the reasoning challenge aligned with the capabilities of modern LLMs. To this end, we introduce Ref-Adv, a modern REC benchmark that avoids short reasoning paths and imposes both reasoning and grounding challenges on contemporary MLLMs. To validate and ensure the quality of the benchmark, we conduct comprehensive in depth ablation studies in section 2 to explore what makes a rigorous modern REC benchmark and compare its reasoning and grounding difficulty with RefCOCO(+/g). Lastly, in section 3, we evaluate 13 contemporary MLLMs on Ref-Adv, both closed source and open source. We report changes in performance with and without Chain-of-Thought (CoT) (Wei et al., 2022) and provide in depth analyses. We believe these results demonstrate the value of Ref-Adv, offer new insights into the capabilities of current MLLMs, and can help guide future research on visual reasoning and REC tasks.

## 2 THE REF-ADV DATASET

### 2.1 DATA SOURCE

We sample from the validation and test splits of COCO (Lin et al., 2014) and OpenImages v7 (Kuznetsova et al., 2020). We filter the images and only use those with panoptic instance anno-

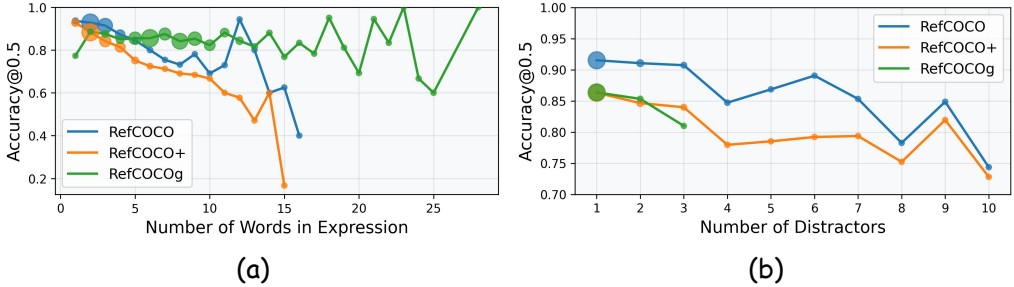

(a)                                        (b)

Figure 2: Accuracy@0.5 (IoU ≥ 0.5) of Qwen on the RefCOCO/+/g validation sets. Marker size is proportional to the number of samples in each bin. (a) is the Acc@0.5 on number of words in expressions, (b) is on distractor count. We can see most cases have short expressions and few distractors.

Table 1: Basic statistics of the validation+test sets of RefCOCO, RefCOCO+, RefCOCOg, and Ref-Adv (Ours). The instance size is represented by its square root. Avg. length: average length of annotations. Vocab.: vocabulary size. Avg. distractors: average number of same category distractors per image. Negation ratio: percentage of expressions using explicit negation.

| Benchmark | Images | Instances | Avg. Length | Avg. Distractors | Negation Ratio | Instance Size | Vocab. |
|---|---|---|---|---|---|---|---|
| RefCOCO 2014 | 3,000 | 7,596 | 3.6 | 3.99 | 0.99% | 105–607 | 3,525 |
| RefCOCO+ 2016 | 3,000 | 7,578 | 3.6 | 3.96 | 3.36% | 105–607 | 4,387 |
| RefCOCOg 2016 | 3,900 | 7,596 | 8.4 | 1.64 | 1.41% | 83–610 | 5,050 |
| Ref-Adv (Ours) | 2,833 | 5,000 | 11.5 | 4.01 | 21.25% | 30-607 | 5,308 |

tations, since this is important for our later pipeline. For the bounding box annotations, we convert all to using the absolute coordinates in the format of [x1, y1, x2, y2]. The input for our data pipeline is the image, the bounding box annotations and category name of each instance, and we will output the referring expression paired with the target instance.

## 2.2 COLLECTION GUIDELINES

As shown in Figure 1, we aim to collect referring data that requires visual reasoning, avoids shortcut solutions, and challenges models. Based on these observations, we propose the following guidelines to mitigate these limitations and yield cases requiring advanced reasoning.

**Distractor Pressure** Distractors are instances of the same category as the target but different instances. To avoid easy grounding based solely on the target category, we select images that have at least 3 candidate instances of the same category as the target, based on the instance annotations of each dataset.

**Language Complexity** RefCOCO(+/g) has an average expression length of around 3 words, which limits language complexity and requires much less visual reasoning. Meanwhile, fixed templates that extract referring information from scene graphs limit diversity in the referring expressions. Therefore, we employ LLMs (e.g., GPT-4o) with carefully designed pipelines to generate more natural and diverse referring expressions while maintaining linguistic complexity.

**Hard Distractors** Simply increasing the number of distractors and the length of the referring expression does not necessarily make the task more challenging because of the "grounding shortcut" illustrated in Figure 1 (c). To reduce such shortcuts (i.e., reliance on redundant descriptors), we ensure the presence of "hard distractors" in the images, defined as distractors that partially match, but do not exactly satisfy, the referring expression. Identifying such pairs and composing expressions around them is central to our data collection process.

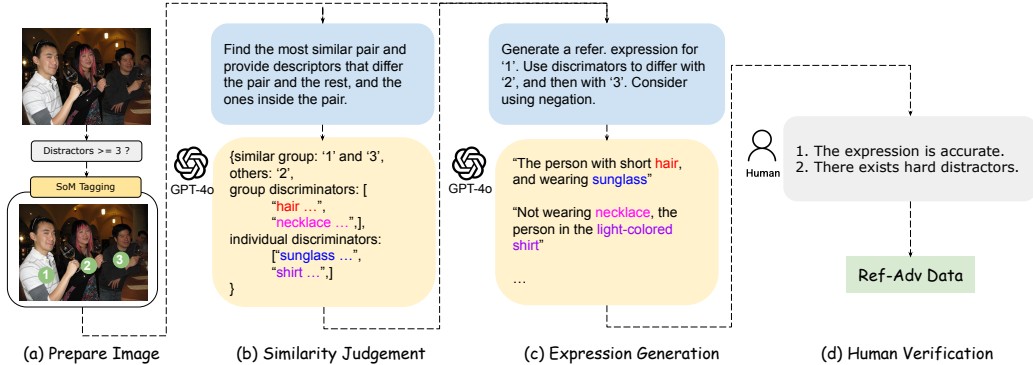

Figure 3: LLM-authored data curation pipeline for Ref-Adv. (a) Prepare Image: filter images, ensure $\geq$ 3 distractors, and add number tags to candidate instances. (b) Similarity Judgement: use GPT-4o to identify the most similar pair and elicit group-level and instance-level discriminators. (c) Expression Generation: compose minimally sufficient referring expressions using discriminators and optional negation. (d) Human Verification: verify expression accuracy and confirm the existence of hard distractors before inclusion.

**Manual Check**   It is laborious and time-consuming to manually select images with hard distractors and generate the referring expressions, so we use LLMs to assist generation. However, LLMs can make mistakes or hallucinate. To ensure accuracy, we perform a human verification pass to confirm the existence of hard distractors and the correctness and unambiguity of the referring expression.

## 2.3   REFERRING EXPRESSION GENERATION PROCESS

As shown in Figure 3, the whole generation process is conducted in four stages. The prompts we use are provided in section 5.

**Input Preparation**   We first filter the images to only keep those with at least 3 candidate instances. We then put a number tag on each instance, similar to Set-of-Marks (Yang et al., 2023), but since we already have instance annotations, we only need to add the number tag to the candidate instances.

### 2.3.1   LLM-AUTHORED PIPELINE

Before detailing the pipeline, we note an important design choice. We first attempted single step prompting of GPT-4o to directly produce complete referring expressions from the image and candidate instances. In practice, GPT-4o frequently produced overspecified descriptions with many redundant descriptors, which enabled shortcut grounding and weakened the need to understand the whole expression. To avoid this behavior, we adopt a two stage procedure: we first elicit discriminative attributes (between group A and group B and within group A), and then compose the final expression from a minimal yet sufficient subset of those attributes.

**Similarity Judgement**   If there is a hard distractor and a target instance, they will be similar in some ways. To encourage the LLMs to identify any such similar pair in the image, we define two groups, group A and group B, where group A contains the hard distractor and the target instance, and group B contains the other distractors. We then prompt the LLMs to identify the two groups and to describe (1) attributes that distinguish the groups and (2) attributes that distinguish the two instances within group A. We ask for multiple alternative descriptions for each distinction. This could help us generate multiple diverse referring expressions for one image and allow us to select the high quality ones.

**Referring Expression Generation**   After the similarity judgement, we obtain a list of paired descriptors that distinguish (1) group A from group B and (2) the two instances within group A. To ensure naturalness and diversity in phrasing, we prompt LLMs to compose referring expressions from combinations of these descriptors. Specifically, we use two alternative strategies: (1) employ the target's descriptors and (2) use the negation of the hard distractor's descriptors. This promotes more diverse and natural expressions. We also explicitly instruct the LLMs to not include number

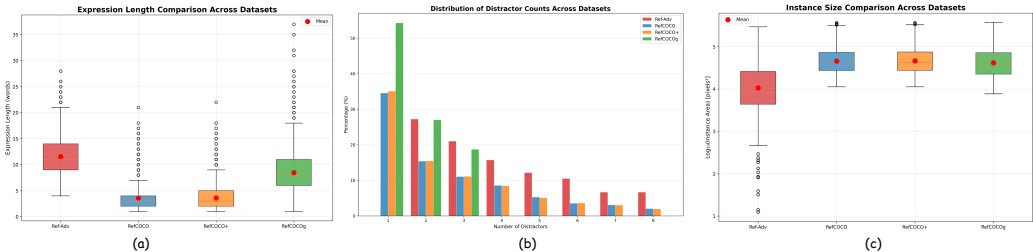

Figure 4: Dataset statistics across REC benchmarks. (a) Expression length comparison. (b) Distribution of distractor counts. (c) Instance size on a log area scale.

tag related descriptions. Although the elicited descriptors alone are sufficient for generation, we find that including the image input at this stage yields more diverse and accurate expressions, so we include the image. After this stage, we obtain multiple candidate referring expressions for each target instance.

### 2.3.2 HUMAN-AUTHORED PIPELINE

We also collect a subset of human-authored referring expressions. For each filtered image, annotators first confirm whether there is a hard distractor pair and, if so, write a referring expression for it. Annotators are instructed to produce diverse and natural phrasing.

### 2.3.3 VERIFICATION PROTOCOL

We verify each image–text pair. Three annotators answer two questions: (1) whether the expression is correct and unambiguous and (2) whether hard distractors are present in the image. Annotators first attempt grounding on the original image (without number tags) using the LLM generated expression. We then show the ground truth box overlaid on the image for reference, allowing reflection if their initial grounding was incorrect. Afterward, annotators record their final decisions on correctness/unambiguity and on the presence of hard distractors. Pairs are presented in a random order per annotator, and a pair is kept only if all three annotators agree. The keep rate is 18.7% for LLM-authored expressions.

### 2.4 QUALITY ANALYSIS

Despite verification to ensure correctness, there remain potential issues for an REC benchmark that could affect fairness and the evaluation of reasoning skills. To further assess the quality of our data, we conduct the following analyses.

**Statistics** As shown in Figure 4 and table 1, Ref-Adv exhibits clear advantages in expression length, vocabulary size, distractor counts, and the negation ratio.

**Model Bias Test** Inspired by Cirik et al. (2018); Chen et al. (2020), we conduct a bias test of modern MLLMs (Qwen2.5-VL-72B and InternVL-3) on RefCOCO(+/g) and Ref-Adv. Here, bias refers to statistical regularities that may arise if training data comes from the same source as an evaluation benchmark, which can benefit performance. We design the test as follows: we replace the referring expression with a fixed prompt ("the one"), keep the same image, and prompt the model to output a bounding box. This test reveals whether model bias helps localize the target. The results are shown in table 2. They suggest that Ref-Adv is less affected by this bias than other benchmarks.

**Textual Reasoning Necessity Test** Prior work (Akula et al., 2020) shows that shuffling word order in RefCOCOg often leaves performance largely intact, indicating weak necessity for textual reasoning in prior REC benchmarks. This lack of degradation could stem from two factors: (1) expressions that only mention the target (or its parts) without referencing distractors and (2) images with no or very few distractors. Both factors reduce the reasoning demand in REC. To validate that Ref-Adv requires reasoning, we extend the test to RefCOCO(+/g) and Ref-Adv for comparison. Rather than shuffling while preserving meaning, we propose a simpler test: we convert the expression to a bag of words and randomize its order in the prompt (e.g., "a red ball with yellow stripes" becomes

Table 2: Accuracy@0.5 after replacing the original referring expressions with the fixed "the one" prompt. $\Delta$ is Fixed@0.5 minus Ref-Adv Fixed@0.5 (shown in blue). With fixed prompt, models achieve higher accuracy on RefCOCO, RefCOCO+, and RefCOCOg than Ref-Adv.

| | RefCOCO | | RefCOCO+ | | RefCOCOg | | Ref-Adv |
| Model | Fixed@0.5 | $\Delta$ vs Ref-Adv | Fixed@0.5 | $\Delta$ vs Ref-Adv | Fixed@0.5 | $\Delta$ vs Ref-Adv | Fixed@0.5 |
|---|---|---|---|---|---|---|---|
| Qwen2.5-VL-72B | 35.1% | +13.7% | 39.4% | +18.0% | 38.3% | +16.9% | 21.4% |
| InternVL-3-14B | 35.9% | +13.1% | 38.0% | +15.2% | 38.2% | +15.4% | 22.8% |

Table 3: Bag-of-words ablation on RefCOCO, RefCOCO+, RefCOCOg, and Ref-Adv. Acc@0.5 with original expressions vs bag-of-words (word order removed). $\Delta$ denotes (BoW $-$ Original).

| | RefCOCO | | | RefCOCO+ | | | RefCOCOg | | | Ref-Adv | | |
| Model | Orig@0.5 | BoW@0.5 | $\Delta$ | Orig@0.5 | BoW@0.5 | $\Delta$ | Orig@0.5 | BoW@0.5 | $\Delta$ | Orig@0.5 | BoW@0.5 | $\Delta$ |
|---|---|---|---|---|---|---|---|---|---|---|---|---|
| Qwen2.5-VL-72B | 92.7% | 82.8% | -9.9% | 88.9% | 78.2% | -10.7% | 89.9% | 75.3% | -14.6% | 58.3% | 41.5% | -16.8% |
| InternVL-3-14B | 92.0% | 84.7% | -7.3% | 87.6% | 81.0% | -6.6% | 88.5% | 74.9% | -13.6% | 52.3% | 38.6% | -13.7% |

"with yellow red ball stripes a"). We evaluate Qwen2.5-VL-72B and InternVL-3 under this setting. Results are shown in table 3, indicating that Ref-Adv indeed requires texual understan and reasoning follow the referring expression exactly.

**Avoidance of "Grounding Shortcut"** As illustrated in Figure 1, RefCOCO(+/g) admits a "grounding shortcut," where a model can localize the target by checking a small subset of descriptors, without reasoning over the entire expression. To validate that Ref-Adv avoids this shortcut, we conduct a ***descriptor-deletion sufficiency*** test. For a given referring expression, we first use Qwen2.5-72B (Yang et al., 2024) to extract all descriptors, randomly delete one, and ask Qwen2.5-72B to rewrite the expression with that descriptor removed. We then evaluate MLLMs on the modified image–text pair. If deleting a descriptor does not affect performance, the descriptor is unnecessary, suggesting a shortcut that succeeds without understanding the full expression. Such shortcuts are exacerbated in datasets with imbalanced numbers of descriptors and distractors. Results are shown in table 4, indicating that Ref-Adv has far fewer grounding shortcuts than others.

## 3 EXPERIMENT

### 3.1 EVALUATION SETUP

**Evaluated Models** We evaluate contemporary state of the art MLLMs, both closed source and open source, on Ref-Adv. The suite includes Qwen2.5-VL series (Bai et al., 2025), InternVL-3 series (Zhu et al., 2025), Gemini 2.5-Flash (Google, 2025a), Gemini 2.5-Pro (Google, 2025b), CogVLM-Grounding (Hong et al., 2024), GLM-4.5V (Team et al., 2025b), GPT-4o (OpenAI, 2024), and Claude-3.5 Sonnet (Anthropic, 2024).

**Evaluation Methods** Set-of-Marks (SoM) overlays numbered marks on candidate objects in the image and leverages a specialized segmenter to provide fine-grained localization, avoiding the need for the MLLM to perform grounding itself. Because GPT-4o and Claude-3.5 have limited grounding ability, we evaluate them using SoM (Yang et al., 2023) with Semantic-SAM (Li et al., 2023). We use Semantic-SAM due to its strong performance on COCO images, one of the sources of Ref-Adv.

For each model (except CogVLM-Grounding which does not support CoT), we evaluate both with and without Chain-of-Thought (CoT). While CoT is uncommon in classic REC benchmark evaluation, Ref-Adv requires more reasoning, so we include CoT in our setup. Table 7 and table 5 report results on Ref-Adv and RefCOCO(+/g) with and without CoT.

**Evaluation Prompts** Models differ in prompt format and output conventions. For example, Qwen2.5-VL-72B uses absolute coordinates, while others use normalized coordinates; CogVLM-Grounding requires the question to strictly follow the form "Where is the 'referring expression'?" to output boxes. To ensure fairness, we adopt best-practice prompts for each model.

Table 4: One descriptor deletion ablation on RefCOCO, RefCOCO+, RefCOCOg, and Ref-Adv. Acc@0.5 with original expressions vs one descriptor deletion (removing a single descriptor in expression). $\Delta$ denotes (1-Desc $-$ Original).

| Model | RefCOCO | | | RefCOCO+ | | | RefCOCOg | | | Ref-Adv | | |
|---|---|---|---|---|---|---|---|---|---|---|---|---|
| | Orig@0.5 | 1D@0.5 | $\Delta$ | Orig@0.5 | 1D@0.5 | $\Delta$ | Orig@0.5 | 1D@0.5 | $\Delta$ | Orig@0.5 | 1D@0.5 | $\Delta$ |
| Qwen2.5-VL-72B | 92.7% | 88.0% | -4.7% | 88.9% | 83.6% | -5.3% | 89.9% | 85.3% | -4.6% | 58.3% | 51.9% | -6.4% |
| InternVL-3-14B | 92.0% | 87.1% | -4.9% | 87.6% | 82.4% | -5.2% | 88.5% | 83.8% | -4.7% | 52.3% | 45.2% | -7.1% |

Table 5: RefCOCO(+/g) and Ref-Adv Acc@0.5 with and without Chain-of-Thought (CoT). $\Delta$ denotes (CoT $-$ Direct).

| Model | RefCOCO | | | RefCOCO+ | | | RefCOCOg | | | Ref-Adv | | |
|---|---|---|---|---|---|---|---|---|---|---|---|---|
| | Direct | CoT | $\Delta$ | Direct | CoT | $\Delta$ | Direct | CoT | $\Delta$ | Direct | CoT | $\Delta$ |
| Qwen2.5-VL-72B | 92.7% | 89.3% | -3.4% | 88.9% | 86.2% | -2.7% | 89.9% | 88.5% | -1.4% | 54.1% | 58.3% | +4.2% |
| InternVL-3-14B | 92.0% | 89.2% | -2.8% | 87.4% | 85.8% | -1.6% | 88.6% | 87.4% | -1.2% | 50.5% | 52.3% | +1.8% |

## 3.2 EVALUATION METRICS

Accuracy serves as a widely adopted metric for evaluating existing REC models. A referring expression instance is deemed successfully grounded when the Intersection over Union (IoU) between the predicted bounding box and the ground truth annotation surpasses 0.5. This conventional evaluation metric is designated as Acc0.5. Here, we implement multiple evaluation protocols, i.e., Accuracy computed under different IoU thresholds such as Acc0.5, Acc0.75, Acc0.9, and mean Accuracy (mAcc) across different IoU criteria, to thoroughly evaluate the precision and robustness.

## 3.3 ANALYSIS

**Effect of Distractor Count** In Ref-Adv, each expression is paired with at least 2 same-category distractors, and images contain roughly 4 distractors on average. Compared with overall Acc0.5, most models show a modest change in the 4–6 group but a larger drop in the $\geq 7$ group (e.g., Qwen2.5-VL-72B+CoT: $-0.2$ and $-2.7$). This trend indicates that handling larger numbers of similar distractors remains a key challenge for current MLLMs.

**Effect of CoT** Table 7 and table 5 show that CoT generally improves performance on Ref-Adv, while it can reduce accuracy on RefCOCO(+/g). We attribute the improvement on Ref-Adv to its heavier reasoning demand; for RefCOCO(+/g), where grounding can often succeed without extensive reasoning, CoT may introduce unnecessary verbosity or error.

It is worth noting that while Argus (Man et al., 2025) reports sizable CoT gains on RefCOCO, its CoT ablations are conducted on VQA style benchmarks by augmenting training with additional CoT data, whereas our study uses off the shelf checkpoints and evaluates directly on RefCOCO(+/g) and Ref-Adv without extra training. RefCOCO(+/g) also contains many short expressions with few distractors, so CoT is often unnecessary and can even harm performance. Moreover, standard multimodal evaluation toolkits such as open compass and VLMEvalKit do not enable CoT for Ref-COCO(+/g), which is consistent with our finding that CoT brings limited benefit in this setting and is more helpful on Ref-Adv, where reasoning demand is higher. This observation is in line with the recent study *"To Think or Not To Think: A Study of Thinking in Rule-Based Visual Reinforcement Fine-Tuning"* (Li et al., 2025), which also reports limited CoT benefits on RefCOCO(+/g).

**Main Results** Table 7 summarizes results on Ref-Adv. With SoM, GPT-4o attains the best performance on Ref-Adv under CoT, suggesting strong reasoning and visual perception capabilities. While other models perform well on RefCOCO(+/g), their accuracy drops markedly on Ref-Adv, revealing gaps in visual reasoning and perception.

**Qualitative Analysis** Figure 5 shows qualitative examples for Qwen2.5-VL-72B and Gemini 2.5-Flash, both with and without CoT. With explicit reasoning, models often follow the intended chain, but in harder cases they fail partway due to incorrect visual perception or a misunderstanding of the referring expression. Notably, models often select the hard distractor as the answer, which indicates that Ref-Adv challenges models to both deeply understand referring expressions and perform accurate visual perception. This suggests that Ref-Adv stresses advanced reasoning and visual perception, and that current state of the art MLLMs still exhibit clear gaps.

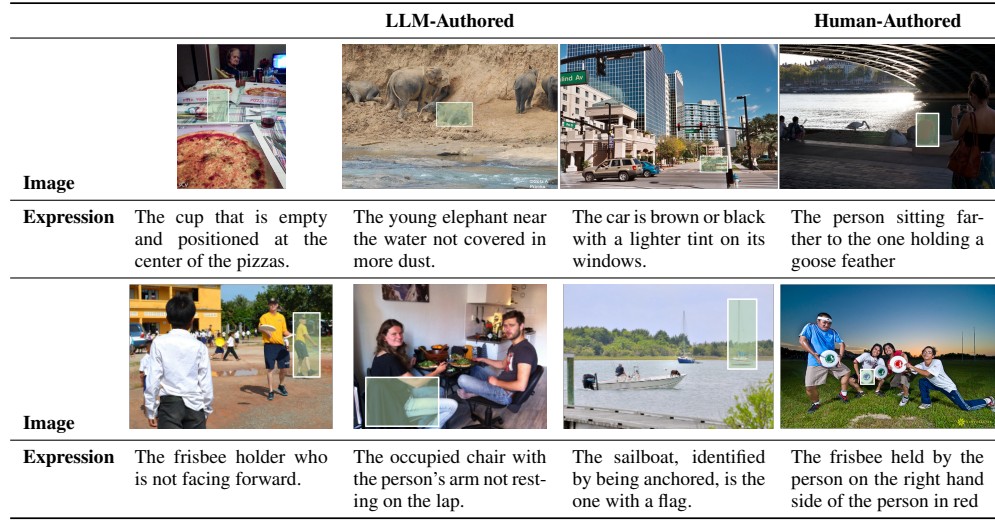

| | LLM-Authored | | | Human-Authored |
|---|---|---|---|---|
| **Image** | | | | |
| **Expression** | The cup that is empty and positioned at the center of the pizzas. | The young elephant near the water not covered in more dust. | The car is brown or black with a lighter tint on its windows. | The person sitting farther to the one holding a goose feather |
| **Image** | | | | |
| **Expression** | The frisbee holder who is not facing forward. | The occupied chair with the person's arm not resting on the lap. | The sailboat, identified by being anchored, is the one with a flag. | The frisbee held by the person on the right hand side of the person in red |

Table 6: Examples from Ref-Adv. Columns 1 to 3 are LLM generated; column 4 is human authored.

Table 7: Main results on Ref-Adv. Rows list models; columns report accuracy at IoU thresholds 0.5, 0.75, and 0.9, and mean accuracy (mAcc). For distractor groups (4–6 and ≥7), we report Acc0.5 and the delta relative to overall Acc0.5.

| Model | Setting | | Acc0.5 | Acc0.75 | Acc0.9 | mAcc | Distractors (Acc0.5) | | | |
|---|---|---|---|---|---|---|---|---|---|---|
| | CoT? | SoM? | | | | | 4–6 | Δ | ≥7 | Δ |
| GPT-4o 2024 | ✗ | ✓ | 52.3 | 31.2 | 13.4 | 27.8 | 53.4 | +1.1 | 51.7 | -0.6 |
| GPT-4o 2024 | ✓ | ✓ | 63.7 | 38.4 | 19.7 | 34.1 | 62.9 | -0.8 | 60.5 | -3.2 |
| Claude-3.5 Sonnet 2024 | ✗ | ✓ | 40.8 | 22.1 | 3.8 | 22.4 | 39.0 | -1.8 | 37.4 | -3.4 |
| Claude-3.5 Sonnet 2024 | ✓ | ✓ | 45.2 | 19.8 | 2.1 | 23.3 | 44.2 | -1.0 | 42.3 | -2.9 |
| Gemini 2.5-Flash 2025a | ✗ | ✗ | 50.6 | 23.7 | 6.9 | 19.2 | 49.5 | -1.1 | 48.9 | -1.7 |
| Gemini 2.5-Flash 2025a | ✓ | ✗ | 59.4 | 35.1 | 16.3 | 30.6 | 58.1 | -1.3 | 55.6 | -3.8 |
| Gemini 2.5-Pro 2025b | ✗ | ✗ | 51.9 | 28.4 | 11.7 | 23.7 | 50.3 | -1.6 | 49.7 | -2.2 |
| Gemini 2.5-Pro 2025b | ✓ | ✗ | 59.1 | 32.6 | 14.2 | 28.3 | 58.0 | -1.1 | 55.9 | -3.2 |
| InternVL-3-7B 2025 | ✗ | ✗ | 49.5 | 39.2 | 21.4 | 33.1 | 49.2 | -0.3 | 48.6 | -0.9 |
| InternVL-3-7B 2025 | ✓ | ✗ | 48.7 | 37.9 | 20.1 | 31.8 | 47.5 | -1.2 | 45.8 | -2.9 |
| InternVL-3-14B 2025 | ✗ | ✗ | 50.5 | 40.7 | 22.8 | 34.2 | 49.7 | -0.8 | 50.3 | -0.2 |
| InternVL-3-14B 2025 | ✓ | ✗ | 52.3 | 42.1 | 24.3 | 35.6 | 51.9 | -0.4 | 49.1 | -3.2 |
| InternVL-3-38B 2025 | ✗ | ✗ | 53.8 | 43.5 | 25.7 | 37.1 | 53.4 | -0.4 | 52.9 | -0.9 |
| InternVL-3-38B 2025 | ✓ | ✗ | 57.2 | 46.8 | 28.9 | 40.3 | 56.9 | -0.3 | 54.1 | -3.1 |
| InternVL-3-78B 2025 | ✗ | ✗ | 54.6 | 44.2 | 26.4 | 37.8 | 53.9 | -0.7 | 53.4 | -1.2 |
| InternVL-3-78B 2025 | ✓ | ✗ | 58.4 | 47.9 | 29.6 | 41.2 | 57.2 | -1.2 | 55.4 | -3.0 |
| Qwen2.5-VL-7B 2025 | ✗ | ✗ | 49.3 | 39.0 | 21.2 | 32.9 | 48.4 | -0.9 | 48.1 | -1.2 |
| Qwen2.5-VL-7B 2025 | ✓ | ✗ | 49.1 | 38.8 | 20.9 | 32.7 | 47.6 | -1.5 | 46.0 | -3.1 |
| Qwen2.5-VL-32B 2025 | ✗ | ✗ | 52.7 | 42.4 | 24.6 | 36.0 | 52.5 | -0.2 | 52.0 | -0.7 |
| Qwen2.5-VL-32B 2025 | ✓ | ✗ | 56.8 | 46.5 | 28.7 | 40.1 | 55.8 | -1.0 | 54.3 | -2.5 |
| Qwen2.5-VL-72B 2025 | ✗ | ✗ | 54.1 | 43.8 | 25.9 | 37.4 | 54.1 | +0.0 | 53.6 | -0.5 |
| Qwen2.5-VL-72B 2025 | ✓ | ✗ | 58.3 | 47.8 | 29.5 | 41.1 | 58.1 | -0.2 | 55.6 | -2.7 |
| GLM-4.5V 2025b | ✗ | ✗ | 52.4 | 42.1 | 24.3 | 35.6 | 51.9 | -0.5 | 51.6 | -0.8 |
| GLM-4.5V 2025b | ✓ | ✗ | 56.9 | 46.6 | 28.8 | 40.2 | 55.9 | -1.0 | 54.6 | -2.3 |
| CogVLM-Grounding 2024 | ✗ | ✗ | 51.5 | 41.2 | 23.4 | 35.0 | 52.4 | +0.9 | 50.8 | -0.7 |

# 4 LITERATURE REVIEW

**Referring Expression Benchmarks.** Segmentation based benchmarks constitute a foundational category in computer vision, with numerous datasets spanning diverse domains and applications (Kuznetsova et al., 2020; Lin et al., 2014; Wang et al., 2022; Du et al., 2023; 2025). The field's foundational benchmarks, including the ReferItGame (Kazemzadeh et al., 2014) and the de facto standard RefCOCO suite (RefCOCO/+/g) (Yu et al., 2016; Mao et al., 2016), have been in-

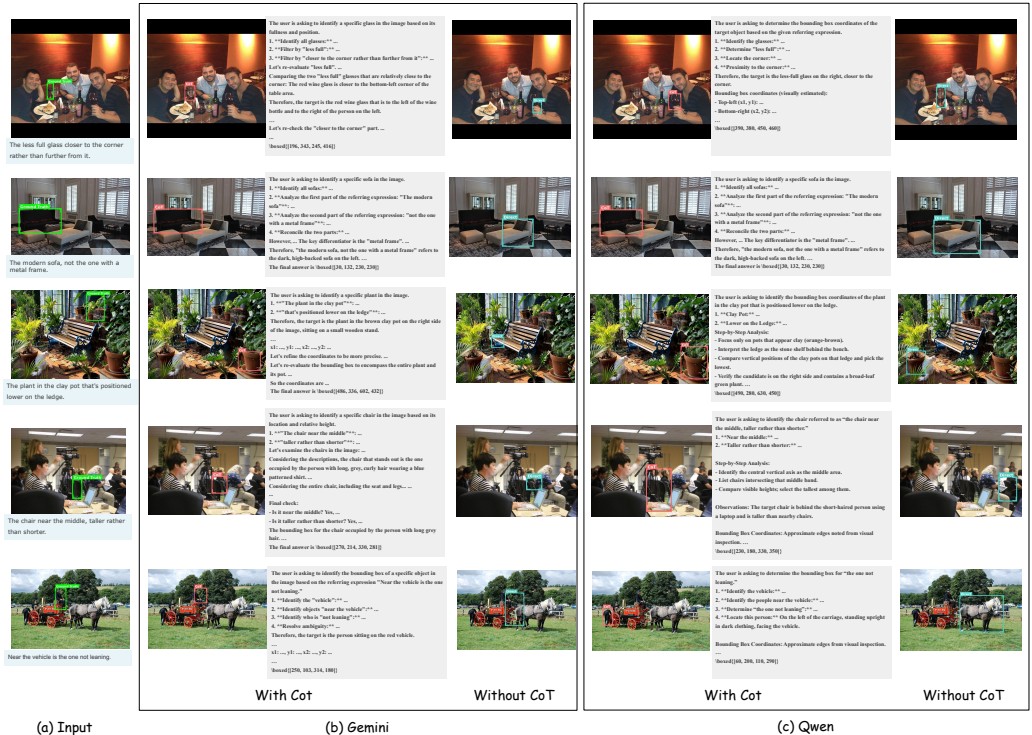

Figure 5: Performance of representative multimodal LLMs on Ref-Adv. We include qualitative examples with and without CoT for Gemini 2.5-Flash and Qwen2.5-VL-72B. CoT answers are shown in a gray box. Hard distractors in Ref-Adv challenge current MLLMs.

strumental in advancing research. However, subsequent analyses revealed that high scores on these datasets can overstate genuine grounding abilities. For example, performance on RefCOCOg often remains high even with shuffled word order, indicating a reliance on superficial cues rather than robust compositional understanding (Akula et al., 2020). To address these cracks in the foundation—namely simplistic expressions and a lack of hard, same-category distractors—a new wave of benchmarks emerged. To directly target reasoning, Cops-Ref (Chen et al., 2020) and its successor FineCops-Ref (Liu et al., 2024) introduced more compositional expressions with explicit distractors and negative examples, while the synthetic CLEVR-Ref+ (Liu et al., 2019) offered a fully controlled environment for diagnostic analysis. Concurrently, other efforts expanded the scope of the REC task itself. gRefCOCO (Liu et al., 2023) introduced multi-target and no-target expressions, PhraseCut (Wu et al., 2020) scaled up to phrase-level segmentation over more categories, and recent works like HC-RefLoCo (Wei et al., 2024) and Ref-L4 (Chen et al., 2024) have pushed for longer, more natural descriptions and corrected label noise in the original benchmarks.

The need for such challenging benchmarks is further amplified by the rapid advancements in Multimodal Large Language Models (MLLMs), which now dominate the field.

**Multimodal Large Language Models.** Recent progress in vision language AI has been driven by large multimodal language models (MLLMs) that combine powerful LLM backbones with vision encoders and alignment tuning for instruction following. A growing body of work has explored visual understanding in LLMs, with grounding ability emerging as an important focus (Bai et al., 2025; Hong et al., 2024; Team et al., 2025b; Lu et al., 2025b;a; 2026). Proprietary models like OpenAI's GPT-4 Vision and Google's Gemini exemplify this trend, while open source counterparts such as Alibaba's Qwen-VL and Shanghai AI Lab's InternVL offer similar capabilities (OpenAI, 2024; Google, 2025a; Bai et al., 2025; Zhu et al., 2025). These systems, trained on massive image text corpora, now achieve near ceiling accuracy (often >90%) on classic referring expression benchmarks (Kazemzadeh et al., 2014; Yu et al., 2016; Mao et al., 2016). However, as the reasoning capabilities of MLLMs rapidly advance, it has become clear that these high scores are insufficient to measure

genuine multi-step reasoning, necessitating an evolution in the REC task itself (Wei et al., 2024; Chen et al., 2024; Dong et al., 2025). This has spurred the development of both more challenging benchmarks and reasoning enhanced models. For example, Moonshot's Kimi-VL (Thinking) applies chain of thought fine tuning and reinforcement learning to strengthen stepwise visual reasoning (Team et al., 2025a), and ZhipuAI's GLM-4.5V explicitly performs step by step grounding to output precise object bounding boxes (Team et al., 2025b). Similarly, new aligned vision language models like CogVLM and DeepSeek-VL2 incorporate mixture of experts or reward optimization to improve visual grounding and coherence, and even commercial chatbots (e.g., Anthropic's Claude 3.5, xAI's Grok) are beginning to integrate advanced multimodal reasoning. Our work builds on these efforts by evaluating a broad suite of state of the art MLLMs—both general purpose and reasoning centric—on a novel REC benchmark designed to stress test their visual grounding and reasoning abilities (Hong et al., 2024; Team et al., 2025b;a; Wu et al., 2024; Anthropic, 2024; xAI, 2025).

## 5 CONCLUSION

In this work, we introduced Ref-Adv, a modern REC benchmark designed to address the reliance on visual shortcuts in existing datasets by requiring genuine multi-step reasoning. We construct Ref-Adv through a two stage pipeline that uses an LLM to compose minimally sufficient referring expressions. Our comprehensive ablation studies (section 2) confirm that Ref-Adv effectively probes both complex textual and visual grounding capabilities. Strikingly, our evaluation of contemporary MLLMs (section 3) revealed a significant performance drop compared to their near-saturated scores on RefCOCO(+/g), exposing a critical overestimation of their visual reasoning abilities. These findings underscore the urgent need for benchmarks that reflect real world visual complexity and offer a clear path forward for developing more robust and capable MLLMs.

## ETHICS STATEMENT

We follow the ICLR Code of Ethics (https://iclr.cc/public/CodeOfEthics). We use large language models to draft candidate expressions and then apply a human verification step with three annotators to ensure correctness and remove ambiguous or unsafe content (Section 2). Annotators worked only with public images and could skip any example. Our benchmark is intended for evaluating grounding and visual reasoning, not for surveillance or biometric identification. We release only expressions, target regions, and dataset identifiers, and we provide usage guidance that discourages applications involving identity inference or sensitive attribute prediction. We are not aware of conflicts of interest.

## REPRODUCIBILITY STATEMENT

Section 2 describes the complete data pipeline, including image sources, filtering with same-class distractors, descriptor elicitation, expression composition, and the three-annotator verification protocol, with a step-by-step diagram in Figure 3. We will release the exact image identifiers, the final referring expressions, target regions, and the JSON schema of our annotations, together with scripts to load and evaluate the data. Evaluation protocols and metrics (Acc0.5/Acc0.75/Acc0.9 and mean Accuracy) are specified in Section 3. To facilitate exact replication, we will provide below artifacts upon publication: (i) the evaluation scripts that compute IoU and accuracy, (ii) the prompts and configuration files for each evaluated model. Together, these artifacts enable end-to-end reproduction of our tables and figures.

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

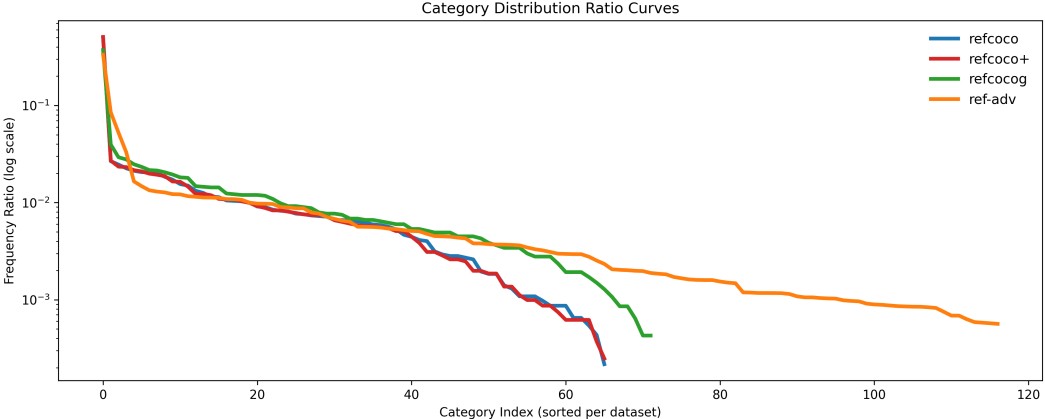

Figure 6: Category distribution ratio curves for RefCOCO, RefCOCO+, RefCOCOg, and Ref-Adv. The frequency ratio is plotted on a logarithmic scale after sorting categories within each dataset.

## A USE OF LLM IN WRITING.

We employed large language models (LLMs) to assist in polishing the text throughout this paper, including refining phrasing, improving clarity, and ensuring grammatical correctness.

## B DATASET CATEGORY DISTRIBUTIONS

Figure 6 visualizes the category level frequency ratios for RefCOCO, RefCOCO+, RefCOCOg, and Ref-Adv on a logarithmic scale after sorting categories within each dataset, and shows that Ref-Adv follows a more long tailed distribution.

## C PROMPT IN DATA COLLECTION

We include the core prompt templates used by our two-stage LLM-authored pipeline described in section 2. Query 1 elicits group-level and intra-pair discriminators; Query 2 composes minimally sufficient referring expressions from those discriminators. Placeholders such as {num_objects} and {target_class} are filled at runtime.

We use structured output in JSON format for the LLMs to ensure the output is in the correct format.

## D LLM API COST FOR DATA COLLECTION

The kept rate is 18.7% for a LLM-authored expression, and each expression will cost about 2300 input tokens and 120 output tokens, with GPT-4o price of \$2.5 per 1M input tokens and \$10 per 1M output tokens, the cost for a LLM-authored expression is $(2300 \times 2.5 + 120 \times 10)/1,000,000 =$ \$0.00695. Given that we need to generate approximately $1/0.187 = 5.35$ expressions to get one kept expression, the effective cost per kept expression is $5.35 \times \$0.00695 = \$0.0372$. For our dataset of 4,000 LLM-authored expressions (others are human-authored), the total cost is approximately $4000 \times 0.0372 = \$148.8$.

```
You are given an image with {num_objects} {target_class} objects labeled
    by integers (1..N).

**Task**:
1) Choose the most similar pair `{{i,j}}` and call that group **A**.
    Everything else is group **B**.
2) Propose exactly **2 group-level discriminators** to separate **A vs B
    **. Each discriminator must have an A-side phrase and a B-side phrase
    .
3) For the two {target_class} objects inside A, propose exactly **4 intra
    -pair discriminators** (2 "noticeable", 2 "unnoticeable"). Each must
    provide a phrase for object `i` and a phrase for object `j`, plus a "
    noticeability" field with value "noticeable" or "unnoticeable".

**Output JSON only**, matching this schema (no extra text):
{{
  "similar_group": {{"ids":[int,int], "label":"A"}},
  "groups": {{"A":[int,...], "B":[int,...]}},
  "group_discriminators":[
    {{"id":"G1","name":string,"A":string,"B":string}},
    {{"id":"G2","name":string,"A":string,"B":string}}
  ],
  "in_pair_discriminators":[
    {{"id":"P1","name":string,"i":string,"j":string,"noticeability":"
        noticeable or unnoticeable"}},
    {{"id":"P2","name":string,"i":string,"j":string,"noticeability":"
        noticeable or unnoticeable"}},
    {{"id":"P3","name":string,"i":string,"j":string,"noticeability":"
        noticeable or unnoticeable"}},
    {{"id":"P4","name":string,"i":string,"j":string,"noticeability":"
        noticeable or unnoticeable"}}
  ]
}}

If the model is multimodal, attend to the image; otherwise rely on the
    provided description/annotations.
```

Listing 1: Query 1: Similarity Judgement and Discriminator Elicitation

```
System: You are a visual assistant that returns JSON only. Follow the
    user's schema exactly. Do not include any extra text.

Image context template: This is an image with {num_objects} {target_class
    }(s) overlaid with integers (1..N).

{image_context}

You are given some observations and a 'target_id'.

**Observations**:
{query1_json}

**Target ID**: {target_id}
**Target Class**: {target_class}

**Task**: Write the referring expressions that refer to {target_class} '
    target_id' based on the observations. Each sentence should use one
    group discriminator (A vs B) and one intra-pair discriminator (
    between the two in A). Return 4 in total.

Return JSON only with this schema:
{{
  "expressions": [
    {{"id":"E1","target_id":int,"group_dids":["G?"],"pair_dids":["P?"],"
        inpair_positive_phrase":string,"inpair_negative_phrase":string,"
        inpair_phrase":"only_positive|only_negative|both","text":string
        }},
    {{"id":"E2","target_id":int,"group_dids":["G?"],"pair_dids":["P?"],"
        inpair_positive_phrase":string,"inpair_negative_phrase":string,"
        inpair_phrase":"only_positive|only_negative|both","text":string
        }},
    {{"id":"E3","target_id":int,"group_dids":["G?"],"pair_dids":["P?"],"
        inpair_positive_phrase":string,"inpair_negative_phrase":string,"
        inpair_phrase":"only_positive|only_negative|both","text":string
        }},
    {{"id":"E4","target_id":int,"group_dids":["G?"],"pair_dids":["P?"],"
        inpair_positive_phrase":string,"inpair_negative_phrase":string,"
        inpair_phrase":"only_positive|only_negative|both","text":string}}
  ]
}}

Explanation example for 'inpair_phrase': if 'inpair_positive_phrase' is "
    sitting" and 'inpair_negative_phrase' is "standing", then "
    only_positive" means "the one sitting"; "only_negative" means "the
    one not standing"; "both" means "the one sitting rather than standing
    ".

Constraints: Use different combinations of group_dids and pair_dids. Vary
     phrasings and sentence structures. Do not mention numeric labels in
     the text.
```

Listing 2: Query 2: Referring Expression Composition

