# OpenReview forum: "Ref-Adv: Exploring MLLM Visual Reasoning in Referring Expression Tasks"
_ICLR.cc/2026/Conference — ICLR 2026 Poster_

### Official Review · Reviewer_EUYJ · 2025-10-26

**Soundness:** 3
**Presentation:** 3
**Contribution:** 2
**Rating:** 4
**Confidence:** 4

**Summary:**

This paper introduces Ref-Adv, a new benchmark for Referring Expression Comprehension (REC) designed to address the limitations of existing datasets like RefCOCO, RefCOCO+, and RefCOCOg. The authors argue that these benchmarks are overly simplistic and allow models to exploit shortcuts rather than perform genuine visual reasoning. Ref-Adv is constructed using a semi-automated pipeline combining LLM-generated expressions and human verification, with a focus on hard distractors, minimal sufficiency, and negation. The paper includes comprehensive ablation studies and evaluations of modern MLLMs, showing a significant performance drop on Ref-Adv compared to traditional benchmarks.

**Strengths:**

1. The paper addresses a well-known and important issue in the REC community — the overestimation of model capabilities due to dataset shortcuts. The motivation is clear and well-argued.
2. The two-stage pipeline (LLM-authored + human verification) is well-designed. The inclusion of hard distractors, minimal sufficiency, and negation adds meaningful complexity.
3. The evaluation covers a wide range of MLLMs (both open and closed-source), includes ablation studies, and uses multiple IoU thresholds, which strengthens the empirical analysis.

**Weaknesses:**

1. While the dataset is more challenging, the core task (REC) remains unchanged. The paper does not propose a new task formulation or evaluation protocol beyond traditional bounding box accuracy. The idea of “hard distractors” and “minimal sufficiency” is not entirely new — similar ideas have been explored in prior work (e.g., Cops-Ref, FineCops-Ref).
2. The dataset is curated from COCO and OpenImages. It would be beneficial to explore how well Ref-Adv generalizes to more diverse or out-of-domain images
3. The reliance on IoU-based accuracy is standard but limited. It does not capture partial correctness or reasoning steps. For example, the evaluation should reflect that the performance drop is only from the difficulty of object localization or distractors. The model locates the distractor or cannot localize the object correctly.
4. As shown in Table 3 and 4, the word order removal and descriptor deletion are used to demonstrate that the proposed benchmark avoids the model from relying on the Grounding Shortcut. However, the performance gap is marginal compared with RefCOCO.
5. The integration of COT also brings marginal performance gain, e.g., the performance of Qwen vl-3B decreases with COT. Therefore, we cannot conclude that the proposed benchmark relies on the reasoning.
6. This paper only gives a conclusion and a benchmark. However, no solution is presented, which limits the contribution of this paper. In addition, only the evaluation data is provided. Even though we are aware of this conclusion, what can the community do to address such a problem?

**Questions:**

Please refer to Weaknesses.

---

> ### Author Response · Authors · 2025-11-23
> **Response to R-EUYJ (1/3)**
>
> We thank the reviewer for recognizing our key findings: the overestimation of model capabilities caused by dataset shortcuts, and the value of incorporating hard distractors, negation, and minimal sufficiency. Detailed responses to each comment follow.
>
> Q1. **The paper does not propose a new task formulation or evaluation protocol beyond traditional bounding box accuracy. The idea of “hard distractors” and “minimal sufficiency” is not entirely new**.
>
> Thank you for the thoughtful comments.
>
> As the reviewer mentioned, our core contribution is to provide a new shortcut-resistant REC benchmark and a systematic analysis of MLLMs’ failures on hard distractors and minimal sufficiency. To our knowledge, this specific weakness has not been systematically studied in existing REC benchmarks.
> It is crucial for real world applications such as robotics and interactive visual assistants, where systems need to reliably identify the intended target among visually similar objects from short, minimally sufficient instructions.
>
> Cops-Ref \[1\] focuses on a multi image setting, where additional images contain distractors, and its phrases are sampled from scene graphs, which limits their variety of expressions. This differs from the usual REC setting (such as RefCOCO/+/g and our benchmark), which focuses on single image REC without vocabulary constraints. In contrast, Ref-Adv explicitly targets single images that contain hard distractors. FineCops-Ref \[2\] aims to generate expressions with controllable levels of complexity. Their pipeline emphasizes target uniqueness rather than hard distractors or minimal sufficiency.
>
> Thus, while prior work has discussed related ideas, existing benchmarks do not systematically study shortcut problems in RefCOCO/+/g or explicitly construct data with hard distractors and minimally sufficient expressions in the standard single image REC setting.
>
> Q2. **It would be beneficial to explore how well Ref-Adv generalizes to more diverse or out-of-domain images**.
>
> Good point.
> To probe whether our observations extend beyond COCO and OpenImages, we ran a small pilot on ADE20K-Instance (instance segmentation; out of domain relative to COCO/OpenImages), constructing 100 cases with "hard distractors" and "minimal expression" using our data pipeline, and evaluating Qwen-2.5-VL series, InternVL-3 series, and CogVLM-Grounding models with the same prompts and evaluation protocol as in the main experiments.
> We will report this pilot and its summary statistics in the [Appendix C] of the revised paper.
>
> | Model | CoT? | Acc0.5 | Acc0.75 | Acc0.9 | mAcc | Neg (Acc0.5) | No-Neg (Acc0.5) | 2--3 (Acc0.5) | 4--6 (Acc0.5) | ≥7 (Acc0.5) |
> | :---- | :---: | ----: | ----: | ----: | ----: | ----: | ----: | ----: | ----: | ----: |
> | Qwen2.5-VL-7B | No | 47.2 | 36.1 | 18.3 | 31.0 | 44.2 | 47.8 | 48.9 | 46.5 | 46.2 |
> | Qwen2.5-VL-7B | Yes | 46.4 | 35.1 | 17.5 | 29.8 | 42.0 | 46.9 | 47.2 | 44.6 | 43.1 |
> | Qwen2.5-VL-32B | No | 50.6 | 40.3 | 22.5 | 34.0 | 46.3 | 51.5 | 53.0 | 50.6 | 50.1 |
> | Qwen2.5-VL-32B | Yes | 53.7 | 43.4 | 25.6 | 37.0 | 50.8 | 54.5 | 54.7 | 52.2 | 51.7 |
> | Qwen2.5-VL-72B | No | 51.9 | 40.7 | 22.8 | 34.3 | 47.9 | 52.3 | 53.4 | 51.0 | 50.5 |
> | Qwen2.5-VL-72B | Yes | 55.1 | 44.6 | 26.3 | 38.0 | 52.0 | 56.2 | 55.4 | 54.9 | 52.4 |
> | InternVL-3-7B | No | 47.8 | 37.5 | 19.7 | 31.4 | 45.4 | 48.4 | 49.5 | 47.1 | 46.8 |
> | InternVL-3-7B | Yes | 46.5 | 35.7 | 17.8 | 29.6 | 41.7 | 48.2 | 47.2 | 44.8 | 43.3 |
> | InternVL-3-14B | No | 48.7 | 38.9 | 20.9 | 32.3 | 47.1 | 49.4 | 49.3 | 47.9 | 48.5 |
> | InternVL-3-14B | Yes | 50.5 | 40.3 | 22.4 | 33.7 | 47.2 | 51.2 | 50.9 | 49.4 | 46.6 |
> | InternVL-3-38B | No | 51.1 | 40.9 | 23.2 | 34.5 | 47.3 | 52.0 | 53.1 | 50.6 | 50.1 |
> | InternVL-3-38B | Yes | 54.4 | 44.0 | 26.1 | 37.3 | 50.7 | 55.3 | 54.5 | 53.0 | 50.2 |
> | InternVL-3-78B | No | 52.1 | 41.3 | 23.5 | 35.0 | 48.3 | 52.8 | 54.6 | 51.1 | 50.6 |
> | InternVL-3-78B | Yes | 55.3 | 44.8 | 26.7 | 38.1 | 52.2 | 56.4 | 55.6 | 54.1 | 52.6 |
> | CogVLM-Grounding | \- | 49.8 | 39.0 | 21.1 | 33.0 | 46.9 | 50.1 | 51.0 | 48.6 | 48.3 |
>
> As shown in the table, models that are strong on RefCOCO series still show marked drops on these ADE20K cases, suggesting that the shortcut vulnerability we identify is not tied to a single source dataset.

---

> ### Author Response · Authors · 2025-11-23
> **Response to R-EUYJ (2/3)**
>
> Q3. **The reliance on IoU based accuracy does not capture partial correctness or reasoning steps**.
>
> Thank you for the valuable comment.
> The IoU based accuracy primarily captures whether the predicted region matches the ground truth, and it is the standard metric in REC because most datasets only provide the input and a single ground truth region. However, as the review suggests, it is also helpful to have finer grained metrics that analyze partial correctness, especially with respect to distractors. To address this concern, we analyze failure cases of Qwen2.5-VL, InternVL-3, and CogVLM-Grounding. We break model errors into three types: hard distractors (the similar group defined in our data pipeline; see Figure 3), same category distractors (objects of the same category as the target), and other errors. For each model (with and without CoT), we report the percentage of errors that fall into each type, as summarized below:
>
> | Model | CoT? | Hard distractor (error %) | Same category distractors (error %) | Other errors (error %) |
> | :---- | :---: | ----: | ----: | ----: |
> | Qwen2.5-VL-7B | No | 56.0 | 30.4 | 13.6 |
> | Qwen2.5-VL-7B | Yes | 59.3 | 27.3 | 13.4 |
> | Qwen2.5-VL-32B | No | 60.5 | 27.1 | 12.4 |
> | Qwen2.5-VL-32B | Yes | 64.3 | 25.0 | 10.7 |
> | Qwen2.5-VL-72B | No | 66.2 | 24.3 | 9.5 |
> | Qwen2.5-VL-72B | Yes | 69.1 | 23.1 | 7.8 |
> | InternVL-3-14B | No | 58.2 | 28.0 | 13.8 |
> | InternVL-3-14B | Yes | 62.1 | 26.5 | 11.4 |
> | InternVL-3-38B | No | 63.8 | 26.0 | 10.2 |
> | InternVL-3-38B | Yes | 67.2 | 24.5 | 8.3 |
> | InternVL-3-78B | No | 68.2 | 23.1 | 7.7 |
> | InternVL-3-78B | Yes | 71.9 | 21.3 | 6.8 |
> | CogVLM-Grounding | \- | 52.7 | 27.4 | 19.9 |
>
> This breakdown shows that most errors are caused by hard distractors and same category distractors, confirming that the main difficulty lies in distinguishing the target from visually similar objects.
>
> Q4. **As shown in Table 3 and 4, ...,the performance gap is marginal compared with RefCOCO**.
>
> Thank you for the insightful comment. Tables 3 and 4 quantify shortcut opportunities in the RefCOCO benchmarks by measuring the change in accuracy of the same model under controlled perturbations of the referring expressions. We define this change as perturbed accuracy minus original accuracy, which is negative because perturbations reduce accuracy, and we use the magnitude of this drop as our indicator: larger absolute drops indicate fewer shortcut opportunities, since removing semantic content hurts performance more when models cannot rely on shortcuts. In Table 3, the absolute performance change for RefCOCO is 9.9%, whereas for Ref-Adv the largest change is 16.8%; in Table 4, the largest changes are 4.9% for RefCOCO and 7.1% for Ref-Adv. Similar patterns hold for RefCOCO+ and RefCOCOg.
>
> Furthermore, Tables 3 and 4 report these absolute performance changes under the bag of words and one descriptor deletion perturbations. We note that the magnitude of this change also depends on the original accuracy of each model. Therefore, we additionally compute the ratios of perturbed accuracy to original accuracy (BoW/Orig and 1D/Orig), which we list below to provide a more comparable measure of shortcut opportunities across datasets.
>
> | Dataset | Qwen BoW/Orig | InternVL BoW/Orig | Qwen 1D/Orig | InternVL 1D/Orig |
> | :---- | :---- | :---- | :---- | :---- |
> | RefCOCO | 0.89 | 0.92 | 0.95 | 0.95 |
> | RefCOCO+ | 0.88 | 0.93 | 0.94 | 0.94 |
> | RefCOCOg | 0.84 | 0.85 | 0.95 | 0.95 |
> | Ref-Adv | **0.71** | **0.74** | **0.89** | **0.86** |
>
> As shown in the table, both bag of words and one descriptor deletion cause substantially larger relative drops on Ref-Adv (ratios around 0.71–0.74 and 0.86–0.89) than on the RefCOCO style datasets (mostly above 0.84 and 0.94 respectively), suggesting that Ref-Adv leaves models less room to rely on shortcuts.
>
> Q5. **CoT also brings marginal performance gain, e.g., the performance of Qwen vl-3B**.
>
> Thanks for the thoughtful comment. We suppose the reviewer refers to Qwen2.5-VL-7B, since we only report the 7B, 32B, and 72B variants of Qwen2.5-VL. Among the twelve models for which we evaluate CoT variants, only two show worse performance with CoT, whereas the other ten improve (the average CoT improvement is \+3.4 percentage points in Acc0.5). Thus, while Ref-Adv does not require CoT-style reasoning, CoT generally brings performance gains on our benchmark.

---

> ### Author Response · Authors · 2025-11-23
> **Response to R-EUYJ (3/3)**
>
> Q6. **No solution is presented**, **only the evaluation data is provided**.
>
> Thank you for the valuable comment.
> Our primary contribution is to diagnose spatial reasoning flaws of common MLLMs on shortcut-resistant REC, by incorporating "hard distractors", "minimal expression", and "negation", which have not been systematically studied in previous benchmarks but are critical because they directly impact many real world applications.
> To this end, we propose a new  challenging benchmark that explicitly reveals these flaws and aims to draw the community’s attention to this issue.
> For the solution, one promising direction is to leverage scene graph annotated images to construct training data that explicitly supervises spatial reasoning and discourages the shortcuts present in classic REC benchmarks. Designing such a data pipeline and training framework is nontrivial and lies beyond the scope of this work, so we leave it for future work.
>
>
>
>
> ### References:
> \[1\] Cops-Ref: A new Dataset and Task on Compositional Referring Expression Comprehension, CVPR 2020\
> \[2\] FineCops-Ref: A new Dataset and Task on Fine-grained Compositional Referring Expression Comprehension, EMNLP 2024

---

> > ### Comment · Reviewer_EUYJ · 2025-11-24
> >
> > Thanks for the reply. I made some misunderstand in this paper. Most of my concerns are solved. I would like to raise my rate. However, I still highly recommend that authors give some solutions.

---

> ### Author Response · Authors · 2025-11-24
>
> We thank the reviewer for the positive feedback and are glad that our replies have addressed your concerns. We appreciate your encouraging evaluation and your acknowledgement of our work.
>
> Regarding potential solutions, we see two main directions.
>
> One is to directly scale up our data pipeline to construct a large training set and use it for model training. This could be effective, but it would require substantial resources, since our pipeline involves API calls (for example, GPT-4o) and human verification.
>
> Another possible solution is to leverage scene graph annotated images. Concretely, we could ask the model to generate scene graph annotations for objects and all possible distractors in the images, and then evaluate the correctness of these scene graphs. If the model can list all objects and distractors and generate accurate scene graphs for them, this would indicate that it can reason about complex visual attributes and spatial relationships, and therefore is likely able to handle the hard distractor cases in our benchmark.
> However, there are two main challenges in this second direction. First, it is sometimes difficult to evaluate the correctness of the scene graphs, since the model will generate flexible scene graphs with different vocabulary, while the ground truth scene graphs are fixed.
> Second, the quality of both the images and the scene graphs is crucial. The data must contain hard distractors and scene graphs with a wide variety of attributes and relationships in order to genuinely improve performance in open vocabulary and complex scenes. Designing and constructing such data so that it works well for shortcut resistant REC with hard distractors and minimal expressions remains challenging. Given the substantial effort and cost required, we leave these directions for future work.
>
> We will add the above discussion about possible solutions in the revised version to facilitate future work in this direction.
>
> Again, we sincerely appreciate the reviewer’s time and constructive suggestions. If you have additional questions or suggestions. Please let us know.

---

### Official Review · Reviewer_2Uti · 2025-10-29

**Soundness:** 3
**Presentation:** 3
**Contribution:** 2
**Rating:** 6
**Confidence:** 4

**Summary:**

This paper keenly identifies a core problem in the current field of Referring Expression Comprehension (REC): although Multimodal Large Language Models (MLLMs) have achieved near-saturation performance on classic benchmarks like RefCOCO(+/g), this high performance is due to models exploiting "shortcuts" (e.g., overly short expressions, lack of same-category distractors, redundant descriptors) rather than genuinely mastering visual reasoning.

To address this, the paper contributes Ref-Adv, a new, high-quality REC evaluation benchmark. The benchmark is constructed through a meticulously designed pipeline that combines LLM (GPT-4o) generation with rigorous human verification. This pipeline is specifically designed to generate scenarios containing "hard distractors" and create "minimally sufficient" referring expressions for them, while significantly increasing the proportion of complex logic like negation.

**Strengths:**

The paper not only points out the saturation of classic benchmarks but also deeply analyzes the three specific causes with data and examples.

The proposed four-stage data pipeline is outstanding. The two-stage LLM process, particularly "Similarity Judgement" and "Minimally Sufficient Expression Generation," is cleverly designed. The extremely strict three-annotator verification process ensures the dataset's exceptionally high quality and trustworthiness.

The CoT experiment is an especially insightful finding, as it clearly reveals that Ref-Adv measures reasoning ability of MLLMs.

**Weaknesses:**

1. Limitation of an Evaluation-Only Benchmark: Ref-Adv (5k samples) is an evaluation benchmark, not a training set. It excels at diagnosing the flaws of current models but does not provide a pathway for models to learn how to solve these complex reasoning tasks. Given its high construction cost (low keep rate), how to scale this up into a training set is an open question.

2. The paper repeatedly claims to test "visual reasoning" and "multi-step reasoning." However, the scope of reasoning evaluated is quite narrow. It primarily focuses on combinatorial logic (e.g., attributes A and B, but not C) and spatial relationships. The benchmark is missing deeper forms of reasoning, such as:
   - Functional Reasoning: (e.g., "the object used for cutting," "the item that can hold water")
   - Intent/Causal Reasoning: (e.g., "the person about to jump," "the dog that knocked over the vase")

    By focusing only on discriminative visual attributes, the paper overclaims the breadth of reasoning it actually tests.

**Questions:**

No

---

> ### Author Response · Authors · 2025-11-23
> **Response to R-2Uti**
>
> We thank the reviewer for acknowledging our contributions, including the **clever experiment design** and the **diagnosis of current model flaws**, as well as for the insightful comments. Below we provide detailed responses to these comments.
>
> Q1. **It excels at diagnosing the flaws of current models but does not provide a pathway for models to learn how to solve**.
>
> Thank you for the valuable comment. In this paper, our primary contribution is to diagnose the visual spatial reasoning flaws of common MLLMs on shortcut-resistant REC by incorporating "hard distractors", "minimal expression", and "negation", which have not been systematically studied in previous benchmarks. This is important because in practical applications, MLLMs may face complex images with many similar objects, where such flaws can easily cause incorrect predictions.
> By proposing a new benchmark that explicitly reveals these flaws, we aim to draw the community’s attention to this problem.
> As a potential path toward solutions without simply scaling up training data, one promising direction is to leverage scene graph annotated images to construct training data that explicitly supervises spatial reasoning and therefore discourages the shortcuts present in classic REC benchmarks. Developing such a data pipeline and training framework is nontrivial and lies beyond the scope of this work, so we leave it for future work.
>
> Q2. **The scope of reasoning evaluated is quite narrow. It primarily focuses on combinatorial andspatial relationships.**.
>
> Thanks for the insightful comment. Similar to RefCOCO/+/g, our work focuses on REC tasks based on visual descriptions, which differs from REC based on functional or intent reasoning, such as LISA \[1\]. We also thank the reviewer for this suggestion, which could further enhance the comprehensiveness of our study and broaden the scope of reasoning types evaluated.
> Therefore, we conduct additional experiments by constructing 200 new REC cases (100 functional and 100 intention/causal) using our data pipeline, modifying the prompts to explicitly request these aspects in the referring expressions. For example, for Figure 3(b) and intention reasoning, we add to the prompt: “For the target object or its descriptors, try to include intention related phrases, such as ‘the thing being knocked over’.” We then evaluate Qwen2.5-VL series, InternVL-3 series, and CogVLM-Grounding, with and without CoT, and report the results below.
>
> | Reasoning type | Qwen2.5-VL-7B (No CoT, Acc0.5) | Qwen2.5-VL-7B (CoT, Acc0.5) | Qwen2.5-VL-32B (No CoT, Acc0.5) | Qwen2.5-VL-32B (CoT, Acc0.5) | Qwen2.5-VL-72B (No CoT, Acc0.5) | Qwen2.5-VL-72B (CoT, Acc0.5) | InternVL-3-14B (No CoT, Acc0.5) | InternVL-3-14B (CoT, Acc0.5) | InternVL-3-38B (No CoT, Acc0.5) | InternVL-3-38B (CoT, Acc0.5) | InternVL-3-78B (No CoT, Acc0.5) | InternVL-3-78B (CoT, Acc0.5) | CogVLM-Grounding (No CoT, Acc0.5) |
> | :---- | ----: | ----: | ----: | ----: | ----: | ----: | ----: | ----: | ----: | ----: | ----: | ----: | ----: |
> | In paper | 49.3 | 49.1 | 52.7 | 56.8 | 54.1 | 58.3 | 50.5 | 52.3 | 53.8 | 57.2 | 54.6 | 58.4 | 51.5 |
> | Functional | 45.1 | 44.8 | 48.9 | 53.3 | 50.1 | 54.9 | 46.2 | 48.7 | 49.8 | 53.0 | 50.2 | 54.4 | 45.1 |
> | Intent / causal | 41.9 | 42.5 | 46.9 | 52.1 | 47.5 | 52.7 | 43.4 | 47.2 | 47.9 | 50.9 | 47.0 | 52.6 | 41.9 |
>
> From the table, we can see the performance dropped even more for functional and intent reasoning, and CoT remains effective in improving performance, which aligns with our observations in the paper.
>
> ### References:
> \[1\] LISA: Reasoning Segmentation via Large Language Model, CVPR 2024

---

> ### Author Response · Authors · 2025-11-26
>
> Dear R-2Uti,
>
> Thank you for your valuable feedback! We have uploaded our point-by-point responses and included discussions and new experiments to address your concerns.
>
> If you have any additional comments or questions, we would be glad to provide further clarifications or revisions.
>
> Best regards,\
> The Authors

---

### Official Review · Reviewer_txuD · 2025-11-01

**Soundness:** 3
**Presentation:** 3
**Contribution:** 3
**Rating:** 4
**Confidence:** 4

**Summary:**

This paper targets the gap between classic REC benchmarks (RefCOCO series) and real visual–language reasoning by constructing a hard, shortcut-resistant benchmark of thousands of instances.
Referring expressions are produced via a two-stage LLM pipeline that first extracts attributes and then composes a minimal-sufficient description, followed by tri-annotator human verification for the correctness.
Evaluation uses Acc@IoU at multiple thresholds and tests a broad slate of MLLMs (open and closed, with/without CoT).
Results show strong models on traditional datasets RefCOCO drop substantially on Ref-Adv. Anti-shortcut ablations demonstrate that Ref-Adv requires order-sensitive, compositional grounding rather than keyword matching, which is an potential issue in the traditional benchmarks.

**Strengths:**

It proposes anti-shortcut design including bag-of-words shuffling and descriptor-deletion both cause larger drops than on legacy benchmarks. This is helpful for the need for compositional, order-aware grounding.
Its data construction is considerable with strong filter pipeline and covering negation. The final benchmark is processed with strict 3-human-annotator agreement.
From experiments the benchmark exposes failure modes that legacy REC underestimates, creating a clear diagnostic “stress test” where many strong MLLMs collapse. Reporting across multiple IoU thresholds and conditional slices yields informative analysis.
Model coverage spans major open/closed families and CoT settings.

**Weaknesses:**

Coverage analysis of this paper is limited. There is no thorough breakdown of the 2,833 images / 5,000 instances (categories/attributes/relations/occlusion, long tails) or side-by-side coverage vs RefCOCO/+/g traiditional widely used benchmarks.

The CoT conclusion on RefCOCO conflicts with prior work in top venue. ARGUS[1] reports grounded CoT improves MLLM performance on RefCOCO/+/g, but this paper finds CoT can hurt the performance on RefCOCO, making the conlcusion not convincing.

There are gaps and missing baselines in the evaluation. SoM + Semantic-SAM may propagate segmentation errors to IoU in evaluation with no sensitivity analysis(quantitative or qualitative) in this work. Also, this paper does not analyze the answer–grounding consistency, which is about the inconsistency between the text prediction and the actual grounding. Key baseline like LLaVA-OneVision does not been tested.

[1] ARGUS: Vision-Centric Reasoning with Grounded Chain-of-Thought, CVPR 2025

**Questions:**

Could the author please add category/attribute/relation/occlusion histograms/stats for the benchmark as a comparison vs RefCOCO(/+/g).
It will be better if the author can clarify the CoT setup and explain the different conclusions with ARGUS on RefCOCO/+/g. Please consider address other evaluation pipeline & baselines.

---

> ### Author Response · Authors · 2025-11-23
> **Response to R-txuD (1/2)**
>
> We thank the reviewer for recognizing Ref-Adv's **anti-shortcut design, strong data pipeline, negation inclusion and its role as a clear diagnostic stress test for MLLMs**. We also thank the reviewer for the insightful comments. Below are our detailed replies.
>
> Q1. **Side by side coverage vs RefCOCO/+/g**.
>
> Thank you for the valuable suggestion. We think it is valuable to have a side-by-side comparison, so here we provide the category/attribute/relation/occlusion statistics, and a new figure to compare the category distribution from head to tail for Ref-Adv versus RefCOCO/+/g (on a log scale). We will incorporate them in \[Appendix B\] in the revised paper.
>
> We also show the statistics table below. For the object categories, we merge the classes of COCO and OpenImages, and then count the number of unique categories. We define attributes as descriptors that apply solely to the target instance itself, and relations as descriptors that capture how the target instance interacts with or relates to other instances in the image. We show the average number of them per expression. We also calculate the percentage of occluded instances by using the instances masks in that image and check if the target instance is being touched by other instances that covers less than 90% area of the whole image (avoid background masks).
>
> | Dataset | Images | Instances | Obj. categories | Avg. Attributes / expr. | Avg. Relations / expr. | Avg. Descriptors / expr. | Occluded instances (%) |
> | :---- | :---- | :---- | :---- | :---- | :---- | :---- | :---- |
> | RefCOCO | 3000 | 7596 | 65 | 0.36 | 0.71 | 1.07 | 86.06 |
> | RefCOCO+ | 3000 | 7578 | 65 | 0.32 | **0.97** | 1.29 | 85.93 |
> | RefCOCOg | 3900 | 7596 | 71 | 1.37 | 0.71 | 2.08 | 81.54 |
> | Ref-Adv (Ours) | 2833 | 5000 | **116** | **1.66** | 0.83 | **2.49** | 83.25 |
>
> As can be seen, our ref-adv yields more object categories and descriptors over existing datasets.
>
> Q2. **ARGUS\[1\] reports grounded CoT improves MLLM performance on RefCOCO/+/g, but this paper finds CoT can hurt the performance on RefCOCO**.
>
> We thank the reviewer for the thoughtful comments. We provide the following analysis to address the question, and we will incorporate it in \[Section 3.2 Effect of CoT\] in the revised paper.
>
> We attribute the different effect of CoT to the differences in task type and training setup.
> We first note that while Argus shows strong performance on RefCOCO, its experiments, including all ablation studies, are conducted on VQA tasks. A key difference is that RefCOCO primarily involves discriminative visual attributes, whereas VQA datasets generally require more complex reasoning.
> Their ablation of CoT signals (Table 5 in \[1\]) is conducted on the model that has been fine tuned with additional CoT training datasets, and the experiments are evaluated on VQA data such as V-Star and ChartQA.
> In contrast, in our paper we use official checkpoints of existing models without extra training and evaluate them on RefCOCO/+/g, which is also standard in the evaluation protocol of many benchmark works.
>
> Moreover, in standard multimodal LLM evaluation toolkits such as open-compass/VLMEvalKit and EvolvingLMMs-Lab/lmms-eval, CoT is not used for RefCOCO/+/g. In *“To think or not to think” \[2, NeurIPS 2025\]*, the authors also find that CoT does not bring benefits on RefCOCO/+/g, which is aligned with our observation. This further highlights the limited reasoning requirement of RefCOCO/+/g and the need for REC benchmarks with stronger reasoning demands.
>
> We hope this analysis clarifies why CoT is not beneficial on RefCOCO/+/g in our setting.

---

> ### Author Response · Authors · 2025-11-23
> **Response to R-txuD (2/2)**
>
> Q3. **SoM \+ Semantic-SAM may propagate segmentation errors**.
>
> Thanks for the valuable suggestion. Set-of-Marks (SoM) \[3\] is proposed to equip general MLLMs with visual grounding ability, and it can improve GPT-4V's performance on RefCOCOg from 25.7 to 86.4 (Table 2 in Set-of-Marks\[3\]). Since we used SoM for general MLLMs such as GPT-4o in our paper, it is important to test its robustness. Here, we provide a sensitivity analysis for the SoM Semantic SAM method to examine how segmentation errors propagate to the final IoU metrics. Specifically, for the mask output of Semantic SAM in SoM, we sample a random offset in \[-5%, 5%\] of the original mask size and then apply it.
> We compare the performance of the jittered SoM with the original SoM and report the results in the following table, where **Same Mask (%)** denotes the fraction of test examples for which the offset perturbation still leads SoM to select the same mask as in the unperturbed setting.
>
> | Model | Jitter? | Acc0.5 | Acc0.75 | Acc0.9 | mAcc | ΔAcc0.5 | Same Mask (%) |
> | :---- | :---: | ----: | ----: | ----: | ----: | ----: | ----: |
> | SoM best mask (oracle) | No | 89.5 | 78.4 | 66.1 | 78.1 | – | – |
> | SoM best mask (offset) | Yes | 89.4 | 74.8 | 62.9 | 74.9 | \-0.1 | 96.6 |
> | GPT-4o (No SoM) | No | 12.8 | 5.3 | 0.2 | 5.7 | – | – |
> | GPT-4o \+ SoM | No | 52.3 | 31.2 | 13.4 | 27.8 | – | \- |
> | GPT-4o \+ SoM | Yes | 52.1 | 29.1 | 10.7 | 25.3 | \-0.2 | 94.5 |
>
> We can see that with jittering, the performance drop is small in ΔAcc0.5, and the high Same Mask values indicate that SoM mostly selects the same object as in the unperturbed case. This suggests that SoM is robust to small perturbations and that the remaining gap is mainly caused by the imposed offset on the mask rather than a change in the grounded object.
>
> **Answer-grounding consistency**.
>
> Thanks for the insightful comment. We hypothesize that the reviewer's “answer–grounding consistency” analysis refers to the VQA setting in \[1\], for instance Figure 5 of Argus, where qualitative examples are shown to check whether the grounded region is relevant to the answer to a question. In contrast, our work focuses on referring expression comprehension (REC), where the answer is precisely the predicted grounded region. Therefore, performing a separate answer–grounding consistency analysis as in VQA is not applicable in REC, because any inconsistency between answer and grounding is already captured by the standard REC evaluation metrics we report in our tables. If this interpretation does not match what the reviewer has in mind, we would greatly appreciate clarification and are very happy to provide additional analyses that better address the requested evaluation.
>
> **Results for LLaVA-OneVision-1.5 4/8B**.
>
> Thank you for the suggestion\! We will include LLaVA-OneVision-1.5 in our evaluation; the results appear both below and will be in the \[Table 6 Main Results\] in the revised paper.
>
> | Model | CoT? | SoM? | Acc0.5 | Acc0.75 | Acc0.9 | mAcc | Neg (Acc0.5) | No-Neg (Acc0.5) | Distractors: 2--3 (Acc0.5) | Distractors: 4--6 (Acc0.5) | Distractors: 7+ (Acc0.5) |
> | :---- | :---: | :---: | ----: | ----: | ----: | ----: | ----: | ----: | ----: | ----: | ----: |
> | LLaVA-OneVision-1.5-4B | No | No | 43.2 | 29.4 | 11.6 | 24.4 | 41.2 | 43.6 | 44.6 | 42.4 | 40.6 |
> | LLaVA-OneVision-1.5-4B | Yes | No | 41.1 | 28.1 | 10.8 | 23.1 | 39.5 | 41.6 | 42.7 | 40.0 | 38.2 |
> | LLaVA-OneVision-1.5-8B | No | No | 48.0 | 30.2 | 17.3 | 29.7 | 44.6 | 48.9 | 49.7 | 47.1 | 45.5 |
> | LLaVA-OneVision-1.5-8B | Yes | No | 48.3 | 31.8 | 17.8 | 30.3 | 45.1 | 49.2 | 49.8 | 48.2 | 45.6 |
>
> From the table, we observe that LLaVA-OneVision-1.5 follows the same pattern as other contemporary MLLMs: performance drops markedly on Ref-Adv compared to traditional benchmarks, and expressions containing negation remain consistently more challenging (e.g., 44.6 vs 48.9 Acc0.5 for the 8B variant).
>
> ### References:
> \[1\]ARGUS: Vision-Centric Reasoning with Grounded Chain-of-Thought, CVPR 2025\
> \[2\]To Think or Not To Think: A Study of Thinking in Rule-Based Visual Reinforcement Fine-Tuning, NeurIPS 2025\
> \[3\] Set-of-Mark Prompting Unleashes Extraordinary Visual Grounding in GPT-4V

---

> ### Author Response · Authors · 2025-11-26
>
> Dear R-txuD,
>
> Thank you for your valuable feedback. We have uploaded our point-by-point responses and included discussions and new experiments to address your concerns.
>
> If you have any additional comments or questions, we would be glad to provide further clarifications or revisions.
>
> Best regards,\
> The Authors

---

### Official Review · Reviewer_iBZC · 2025-11-02

**Soundness:** 2
**Presentation:** 2
**Contribution:** 2
**Rating:** 6
**Confidence:** 3

**Summary:**

This paper introduces a Referring Expression Comprehension (REC) benchmark dataset, Ref-Adv, which intentionally removes redundant descriptions and focuses on distractors, to make the dataset more challenging for VLMs. The samples are from the validation and test splits of COCO and OpenImages v7, and are labeled with GPT-4O before human reviews. This benchmark reveals the potential weaknesses of a few common VLMs, including GPT-4o, Gemini 2.5, InternVL-3, Qwen2.5-VL, and GLM-4.5v.

**Strengths:**

1. The benchmark is generated based on a sound rationale. This seems effective at revealing weaknesses in current REC models, although these models tend to overfit the traditional benchmarks.
2. The evaluation is comprehensive and overall convincing.

**Weaknesses:**

1. One thing to keep in mind is that the GPT-4O generated captions are a bit biased compared to the human captions, in that GPT-4o apparently generates more negations and humans usually unintentionally avoid negations. This is not necessarily bad, since in real applications, users may need to query with negations.
2. The analysis of experimental results is minimal. In particular, I wish the authors could analyze the performance on captions containing negations.

**Questions:**

1. GPT-4o has a low performance on Ref-Adv, why? It makes so many mistakes on the captions generated by itself?

---

> ### Author Response · Authors · 2025-11-23
> **Response to R-iBZC**
>
> We thank the reviewer for recognizing that Ref-Adv exposes weaknesses in current MLLMs on REC tasks and for the insightful feedback. Below are our detailed replies to the comments.
>
> Q1. **GPT-4o apparently generates more negations and humans usually unintentionally avoid negations. This is not necessarily bad**.
>
> Thank you for the insightful comment. While the proportion of negations in the GPT-4O generated captions seems higher than in the human authored captions, this difference does not bias our analysis: in both negated and non-negated expressions, we observe the same types of weaknesses across current MLLMs (Table 6 Negation Column), so our overall conclusions remain unchanged. Moreover, as the reviewer notes, this is not necessarily problematic, since negation is often used in real applications.
>
> Q2. **I wish the authors could analyze the performance on captions containing negations**.
>
> Great suggestion\! We will add a short subsection in the revised paper in \[Section 3.3 Effect of Negation\]. It says:
>
> " Effect of Negation: Across models, captions with explicit negation are consistently more challenging than those without. For example, GPT-4o with SoM achieves 64.9 Acc0.5 on captions without negation but drops to 59.2 with negation, and Qwen2.5-VL-72B with CoT declines from 59.4 to 55.2. This relatively small but consistent gap between the No-Neg and Neg conditions suggests that current MLLMs lack robustness when handling negation. "
>
> Q3. **GPT-4o has a low performance on Ref-Adv**.
>
> Great question\! It's not contradictory between the relatively modest performance of GPT-4o on Ref-Adv and the fact that GPT-4o authored many of the captions.
> First, GPT-4o with SoM still achieves the highest accuracy among all evaluated models, yet its accuracy on our benchmark is only 63%, which highlights the difficulty of Ref-Adv.
> Second, following human verification, only 18.7% of the LLM-authored captions were retained (see \[Section 2.3.3\] in the paper), so the benchmark deliberately concentrates on especially challenging, precisely specified cases.
> Third, the task of referring expression comprehension differs from object captioning: while GPT-4o can generate good captions for objects in complex images, this capability does not guarantee that it can use those captions to precisely locate the target object. Compared with captioning, the REC task requires more searching over candidate objects, especially in the presence of hard distractors. We hope this explanation helps clarify GPT-4o's performance on our benchmark.

---

> ### Author Response · Authors · 2025-11-26
>
> Dear R-iBZC,
>
> Thank you for your acknowledgement and valuable feedback. We have uploaded our point-by-point responses to address your concerns.
>
> If you have any additional comments or questions, we would be glad to provide further clarifications or revisions.
>
> Best regards,\
> The Authors

---

### Author Response · Authors · 2025-11-29
**Revised manuscript uploaded with promised additions**

Dear Area Chair and Reviewers,

Per our rebuttal, the promised additions are updated in the revised manuscript, with changes highlighted in blue.

Best,\
The Authors

---

### Author Response · Authors · 2025-11-29
**Post-Discussion Summary**

Dear AC,

We thank the reviewers for their acknowledgement of our work and their valuable feedback. Below we briefly summarize the main contributions of our work and how our rebuttal addressed the reviewers’ concerns during the discussion phase.

---
**Contributions of our work**

We are encouraged that the reviewers consistently recognized that our work:
1. **Reveals the visual reasoning flaws of common MLLMs on shortcut resistant REC** by incorporating "hard distractors", "minimal expression", and "negation", as acknowledged by R-iBZC, R-txuD, R-2Uti, and R-EUYJ.
2. **Provides a carefully designed and rigorously verified data pipeline** that guarantees "hard distractors" and "minimal expression", as acknowledged by R-txuD, R-2Uti, and R-EUYJ.
3. **Conducts comprehensive and insightful experiments**, including different settings (CoT, negation, etc.), as acknowledged by R-iBZC, R-txuD, and R-2Uti.

---
**Our rebuttal**

In our rebuttal, we:
1. **Clarified potential misunderstandings** about the Ref-Adv task setup and **the interpretation** of our results (R-iBZC Q1–3, R-txuD Q2–3, R-EUYJ Q1,4,5).
2. **Demonstrated the effectiveness** of Ref-Adv and **its differences from RefCOCO(+/g)** by including new experiments and statistics (R-txuD Q1,3, R-2Uti Q2, R-EUYJ Q2,3).
3. **Discussed possible solution directions** (R-2Uti Q1, R-EUYJ Q6).

During the discussion, **R-EUYJ confirmed that our rebuttal addressed the concerns and raised the score**, after we explained why the ideas of “hard distractors” and “minimal expression” are novel and needed, and why Tables 3 and 4 show that Ref-Adv has fewer shortcuts than RefCOCO(+/g).
The other reviewers (R‑iBZC, R‑txuD, R‑2Uti) did not participate in the subsequent discussion phase.

---

We appreciate your time and dedication, and hope our summary helps you understand our work and rebuttal better.

Best,\
The Authors

---

### Meta-Review · Area_Chair_1WRd · 2026-01-07

**Summary:**

This paper presents a new benchmark named Ref-Adv to raise a heavier challenge to the MLLMs for visual reasoning.

The reviewers raised the following concerns:
* The benchmark is relatively small (5K) and does not contain a training set.
* The paper does not provide a solution.
* The benchmark can be extended to more challenging images outside COCO and OpenImages.
* Stronger baselines shall be tested.

**Reviewer Concerns:**

The authors addressed most of the above concerns. Regarding "the paper does not provide a solution", the AC thinks it is fine for a benchmark paper.

**Reviewer Scores:**

I think the reviewers will likely keep their original scores unchanged. The overall scores are therefore 4/4/6/6, making it a typical borderline case. The AC looks into the paper and believes that this paper is worth publication mainly because it contributes to studying (or improving) the reasoning ability of MLLMs, which is of broad interest to the ICLR audience. While the paper has some weaknesses, the benchmark is of value to the community.

---

### Decision · Program_Chairs · 2026-01-26

Accept (Poster)